# A phytoplankton bloom caused by the super cyclonic storm Amphan in the central Bay of Bengal

Haojie Huang[1,4], Linfei Bai[3,4], Hao Shen[4], Xiaoqi Ding[3,4], Rui Wang[3,4], Haibin LÜ[1,2,4]

[1]Jiangsu Key Laboratory of Marine Bioresources and Environment /Jiangsu Key Laboratory of Marine Biotechnology, Jiangsu Ocean University, Lianyungang, Jiangsu province, China
[2]Co-Innovation Center of Jiangsu Marine Bio-industry Technology, Jiangsu Ocean University, Lianyungang, Jiangsu province, China
[3]Lianyungang Meteorological Bureau, Lianyungang, Jiangsu province, China
[4]School of Marine Technology and Geomatics, Jiangsu Ocean University, Lianyungang, Jiangsu province, China

*Correspondence to*: Haibin LÜ (haibin_lv@jou.edu.cn)

**Abstract.** The supercyclonic storm Amphan originated in the central Bay of Bengal (BoB) in May 2020, and a phytoplankton bloom occurred in the upper ocean that was devoid of background nutrients. The dynamic mechanism of the chlorophyll-a (Chl-a) bloom was researched based on reanalysis data, remote sensing and Argo float data. During the passage of Amphan, an inertial oscillation with a two-day period appeared in the thermocline and lasted for approximately two weeks. After the passage of Amphan, a cyclonic eddy with a maximum vorticity of approximately 0.36 s$^{-1}$ formed in the study area (Box A). Additionally, horizontal transport of Chl-a also occurred when the maximum inlet fluxes through the western and northern sides of Box A were 0.304 mg·m$^{-2}$·s$^{-1}$ and -0.199 mg·m$^{-2}$·s$^{-1}$, respectively. With the weakened thermocline and thinner barrier layer thickness (BLT), nitrate and Chl-a were uplifted to the upper ocean by upwelling. Then, with the high photosynthetically available radiation (PAR) in the upper ocean, a phytoplankton bloom occurred. This study provides new insights into the biological responses in the BoB during the passage of tropical cyclones (TCs).

## 1 Introduction

Tropical cyclones (TCs) are serious natural disasters. Heavy rainfall, storm surges and other disasters accompany TCs, resulting in huge economic losses and casualties(Girishkumar and Ravichandran, 2012; Lu et al., 2020). Rapid saltwater intrusion caused by TCs can lead to a range of ecological damage by causing land salinisation and disruption of upstream freshwater system (Mitra, 2020). Soil moisture is critical in climate evolution (Cai et al., 2017), and soil salinisation can alter soil moisture and indirectly affect climate change. Additionally, internal waves have also been observed during and after TC passage, which can exert force and torque on the tendon legs of offshore oil platforms(LÜ et al., 2016). Strong cyclones can also cause deformation of structures such as dams and pose a risk to undersea pipelines (Guan, 2019; Qiu, 2020). The Bay of Bengal (BoB), which has a tropical monsoon climate, is a semienclosed basin in the North India Ocean, which has one of the highest cyclone frequencies in the world(Vinayachandran, 2003). TCs in the BoB appear more frequently during the premonsoon (April to June) and postmonsoon (October to December) periods(Girishkumar and Ravichandran, 2012), which

are three to four times more frequent than TCs in the Arabian Sea(Akter and Tsuboki, 2014). A strong saline stratification forms in the upper ocean due to the influx of freshwater from river discharge and monsoonal rainfall(He, 2020; Thadathil et al., 2007). Generally, TCs can cause phytoplankton blooms, thus increasing the concentration of Chl-a at the sea surface(Nayak Sr, 2001; Rao Kh, 2006; Vinayachandran, 2003; Zhao et al., 2015). During the passage of Phethai(Xia et al., 2022), a cyclonic eddy in eastern Sir Lanka was enhanced with a maximum vorticity of 0.36 $s^{-1}$, which triggered a Chl-a bloom with a maximum Chl-a concentration of 0.6445 mg·$m^{-3}$.

 In other sea areas, after the passage of Typhoon Nuri in the western Pacific(Zhao et al., 2009), nutrient-rich water was transported from the deep layer and the coast to offshore regions, nourishing phytoplankton biomass, and then two Chl-a patches were observed near the Pearl River Estuary and the Dongsha Archipelago. Typhoon Lupit(Cheung et al., 2012) in the northwestern Pacific also caused Chl-a blooms. Some studies have shown that phytoplankton blooms are caused by Ekman pumping(Vinayachandran, 2003), while others have suggested that blooms are caused by cyclonic eddies(Gomes et al., 2000). In the Gulf of Mexico, a tropical sea area in the western Atlantic Ocean, after the passage of Michael in the northeastern Gulf of Mexico (Nyadjro et al., 2022), chlorophyll concentrations also rose by nearly 0.7 mg·$m^{-3}$ significantly.

TC Amphan, which was the first extremely serious storm of this century, intensified from a cyclonic storm (CS) to a super cyclone within 36 hours(Golder et al., 2021). TC Amphan formed over the Indian Ocean on May 5, 2020, and enhanced to a high speed. On May 21, it made a landfall along the coast of Bangladesh. Four days after the passage of Amphan, a phytoplankton bloom occurred in the central Bay of Bengal, where the upper ocean was devoid of background nutrients. Why this phytoplankton bloom happened?  However, little attention has been paid on it in this study area.

In this study, the phytoplankton bloom mechanism in the central BoB caused by TC Amphan was investigated. The data and methodology are provided in Section 2, followed by the results in Section 3, and a discussion is provided in Section 4. Finally, we provide our conclusions in Section 5.

## 2 Data and Methods

### 2.1 Study area

The study area was located in the central BoB. Amphan was the costliest storm ever recorded in the BoB due to its catastrophic damage. We examined one box, named Box A (13.5° N-15.5° N, 86° N-88° E), along the path of Amphan (Fig. 1). First, a depression formed in the southeast BoB at 9.5° N,87.5° E on May 15, which intensified into a deep depression in a short amount of time and further developed into a cyclonic storm the following day. Then, Amphan rapidly intensified into a super cyclonic storm with a maximum wind speed of 74.59 m·$s^{-1}$. Finally, it steadily weakened and made landfall on May 20.

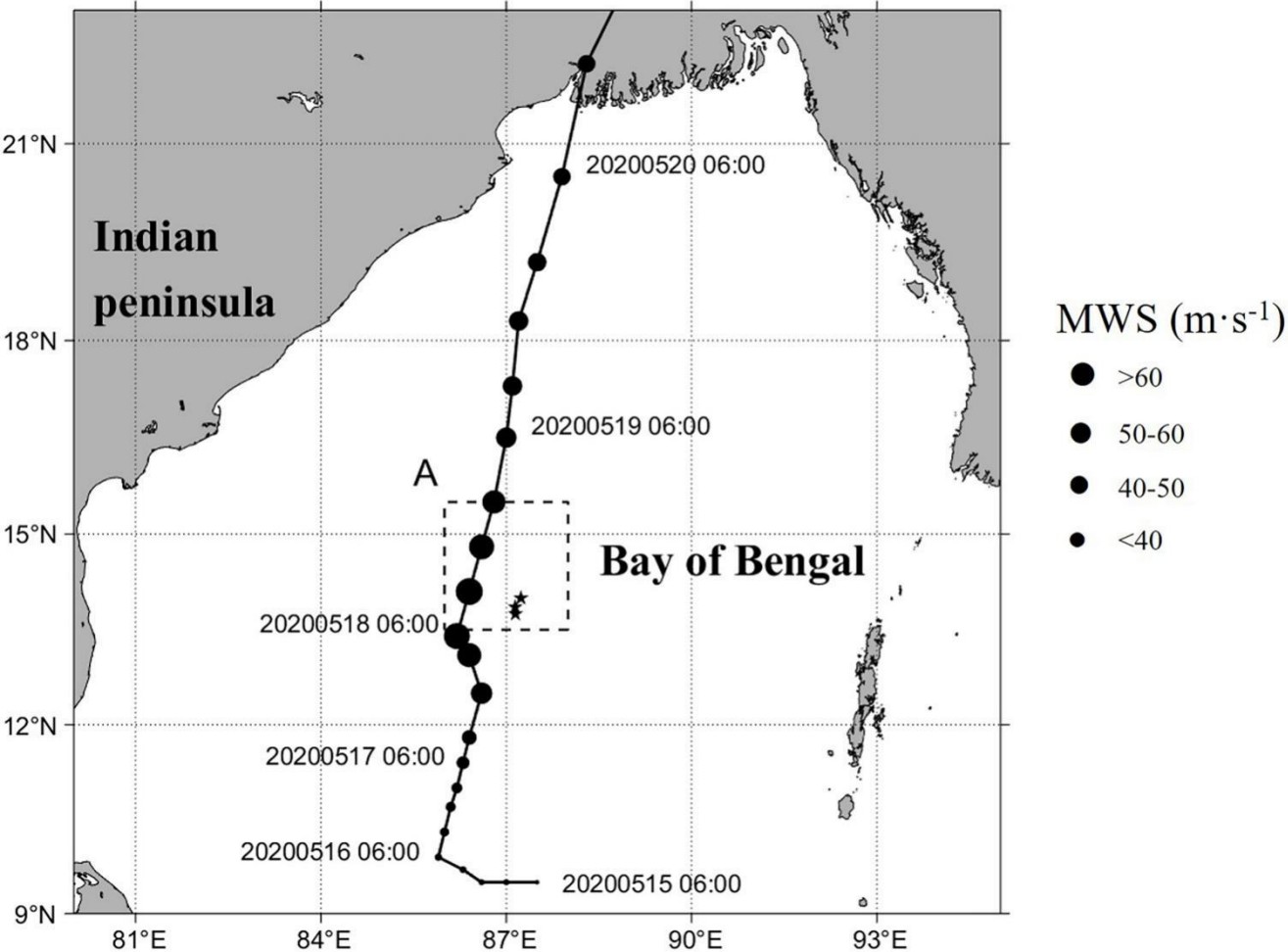

**Figure 1. Track of TC Amphan. TC center locations are marked as black circles, indicated with their time in year-month-date-hour format. Locations of the Argo floats are indicated by stars (2 902 769). The study area of Box A is denoted by the dashed black square: 13.5° N-15.5° N, 86° E-88° E.**

## 2.2 Data

The track of TC Amphan is available from the Joint Typhoon Warning Center (JTWC, https://www.metoc.navy.mil/jtwc/jtwc.html).

Wind vector and sea surface temperature (SST) data are available from remote sensing systems (RSS, http://www.remss.com/measurements/).

High resolution (0.83°×0.83°) daily sea surface height (SSH), sea current velocity, temperature and Chl-a data are available from the Copernicus Marine Environment Monitoring Service (CMEMS, http://marine.copernicus.eu/).

Climatic nitrate data with a resolution of 1°×1° for May are accessible from the World Ocean Atlas (WOA, https://www.ncei.noaa.gov/products/world-ocean-atlas).

Temperature and salinity profiles are accessible from the International Argo Project (http://www.argodatamgt.org/accessed), which began in 1999(Gould, 2005; Jayne et al., 2017; Roemmich et al., 2009). The buoy positions in Fig. 1 are marked by stars. The Argo platform number in this study is 2 902 769.

Daily surface Chl-a concentrations with a 4 km spatial resolution and photosynthetically active radiation (PAR) data are accessible from the satellite remote sensing of the GlobColour Project (https://www.globcolour.info/).

The reanalyzed daily Chl-a data with a spatial resolution of 4 km are accessible at https://data.marine.copernicus.eu/products.

Ocean color remote sensing satellites can be used to observe chlorophyll distribution and phytoplankton blooms due to the different reflectivity of electromagnetic radiation. The chlorophyll content can be estimated by comparing the reflectance at different wavelengths(Balaguru et al., 2012; Dwivedi et al., 2008).

## 2.3 Methods

### 2.3.1 Ekman pumping velocity (EPV)

The vertical mixing of the upper ocean that is caused by TCs is inseparably related to Ekman pumping. Equation (1) and (2) can be used to calculate EPV (Bai et al., 2023; Gaube et al., 2013):

$$W_E = Curl(\vec{\tau}/\rho_0 f) , \tag{1}$$

$$\vec{\tau} = \rho_a C_D |\overrightarrow{U_{10}}|\overrightarrow{U_{10}} , \tag{2}$$

where the Coriolis parameter $f$ is the function of latitude $\theta$ and the earth's rotational velocity $\omega$, which is calculated by $f = 2\omega \sin\theta$, $\vec{\tau}$ is the wind stress, $\rho_0$ and $\rho_a$ represent the seawater density and the air density, respectively, $C_D$ is the drag coefficient and $\overrightarrow{U_{10}}$ represents the wind speed at 10 meters above sea level(Wang et al., 2010).

### 2.3.2 Vorticity

The curl of a sea current vector can be calculated by Equation (3) (Lu et al., 2020):

$$Curl = dv/dx - du/dy , \tag{3}$$

where $u$ and $v$ are the components of sea current velocity along the $x$ and $y$ directions, respectively.

### 2.3.3 Brunt–Vaisala Frequency

The Brunt–Vaisala frequency ($N$) can describe the stability of seawater and the atmosphere(Bai et al., 2023). It can be calculated as follows:

$$N = \sqrt{\frac{-g}{\rho} + \frac{d(\rho)}{d(z)}} , \tag{4}$$

Where $\rho$ is the seawater potential density, $g$ is the gravitational acceleration and $z$ is the vertical coordinate component. The location of the thermocline is defined at the depth of the maximum $N$(Lu et al., 2020).

### 2.3.4 Barrier layer thickness

The mixing layer depth (MLD), isothermal layer depth (ILD) and barrier layer thickness (BLT) in this study area were examined(He, 2020). MLD is the increase in the upper ocean potential density ($\Delta\sigma_\theta$) when the SST decreases by 0.5 °C ($\Delta T = -0.5$ °C) (Equations 5-7), the depth of ILD is the depth (D) at which the temperature is 0.5 °C lower than the SST (Equations 8-9), and the BLT is the difference between the MLD and ILD (Equation 10):

$$\Delta\sigma_\theta = \sigma_{MLD} - \sigma_{10} \, , \tag{5}$$

$$\Delta\sigma_\theta = \sigma_\theta(T_{10} + \Delta T, S_{10}, P_0) - \sigma_\theta(T_{10}, S_{10}, P_0) \, , \tag{6}$$

$$MLD = \sigma_{MLD} \, , \tag{7}$$

$$\Delta T = T_{ILD} - T_{10} \, , \tag{8}$$

$$ILD = D_{ILD} \, , \tag{9}$$

$$BLD = ILD - MLD \, , \tag{10}$$

where $T_{10}$ and $S_{10}$ represent the temperature and salinity at 10 meters' depth, separately, and $P_0$ represents the pressure at the sea surface.

## 3 Results

### 3.1 Amphan's path and wind field

According to the information from India Meteorological Department, TCs can be classified by their near-center maximum wind speed (MWS), ranging from a depression (TD, 31-49 km/h) to a super cyclonic storm (SCS, $\geq$ 220 km/h). Amphan was the first and most serious cyclonic storm in May 2020. It originally appeared at $9.5°\,N, 87.5°\,E$ on May 15, 2020, and landed at $22.2°\,N, 88.3°\,E$ on May 20, 2020 (Table 1). Since Amphan's formation, it generally continued to move northward along the Bay of Bengal.

**Table 1: Information on TC Amphan (MWS: maximum wind speed).**

| Latitude (° $N$) | Longitude (° $E$) | Time | MWS (m $s^{-1}$) | Minimum Sea Level Pressure (mb) |
|---|---|---|---|---|
| 9.5 | 87.0 | 2020/05/15/12 | 15.43 | 1002 |
| 9.7 | 86.3 | 2020/05/16/00 | 18.00 | 999 |
| 10.3 | 86.0 | 2020/05/16/12 | 25.72 | 992 |
| 11.0 | 86.2 | 2020/05/17/00 | 30.86 | 982 |
| 11.8 | 86.4 | 2020/05/17/12 | 41.15 | 970 |
| 13.1 | 86.4 | 2020/05/18/00 | 66.87 | 919 |
| 14.1 | 86.4 | 2020/05/18/12 | 74.59 | 901 |

| 15.5 | 86.8 | 2020/05/19/00 | 64.30 | 925 |
| 17.3 | 87.1 | 2020/05/19/12 | 54.01 | 946 |
| 19.2 | 87.5 | 2020/05/20/00 | 51.44 | 947 |
| 22.2 | 88.3 | 2020/05/20/12 | 48.87 | 954 |
| 25.2 | 90.1 | 2020/05/21/00 | 25.72 | 984 |

### 3.2 Chl-a

125 Bad weather conditions during the TC resulted in the poor usability of daily satellite Chl-a data. Zhao et al. determined the Chl-a concentration by synthesizing data for three days(Zhao et al., 2015). Here, we uses the same method to display the Chl-a changes in the study zone. Fig. 2 displays the sea surface Chl-a concentration composited for May 12-14 and May 25-27 before and after the passage of Amphan. Fig. 3 shows the three-day averaged Chl-a concentration over time. The Chl-a concentration in Box A increased at an extremely high speed from May 22 until it reached a peak of 0.3071 mg/m$^3$, where

the high concentration was maintained for approximately five days, and then it rapidly decreased on May 28. The reasons for the Chl-a bloom are provided below.

Fig. 4 shows the Chl-a fluxes through southern, northern, eastern and western sides of Box A from May 5 to 31. We can calculate the Chl-a fluxes on the southern and northern sides by multiplying the Chl-a concentration by v, and by u achieves the Chl-a fluxes on the eastern and western sides (Xia et al., 2022). Positive values and negative values represent the flow-in

and flow-out on the southern and western sides, respectively; in contrast, the flux-in and flux-out on the eastern and northern sides were exactly the opposite. First, it was clearly shown that Chl-a entered from both the western and the northern sides and discharged from the other two sides. Second, the high Chl-a flux in the upper ocean was maintained for 4 days (Fig. 4b, 4d); the maximum flux from the western side was 0.304 mg·m$^{-2}$·s$^{-1}$, and that on the northern side reached -0.199 mg·m$^{-2}$·s$^{-1}$. Finally, the surface horizontal transport of Chl-a rarely changed after May 22.

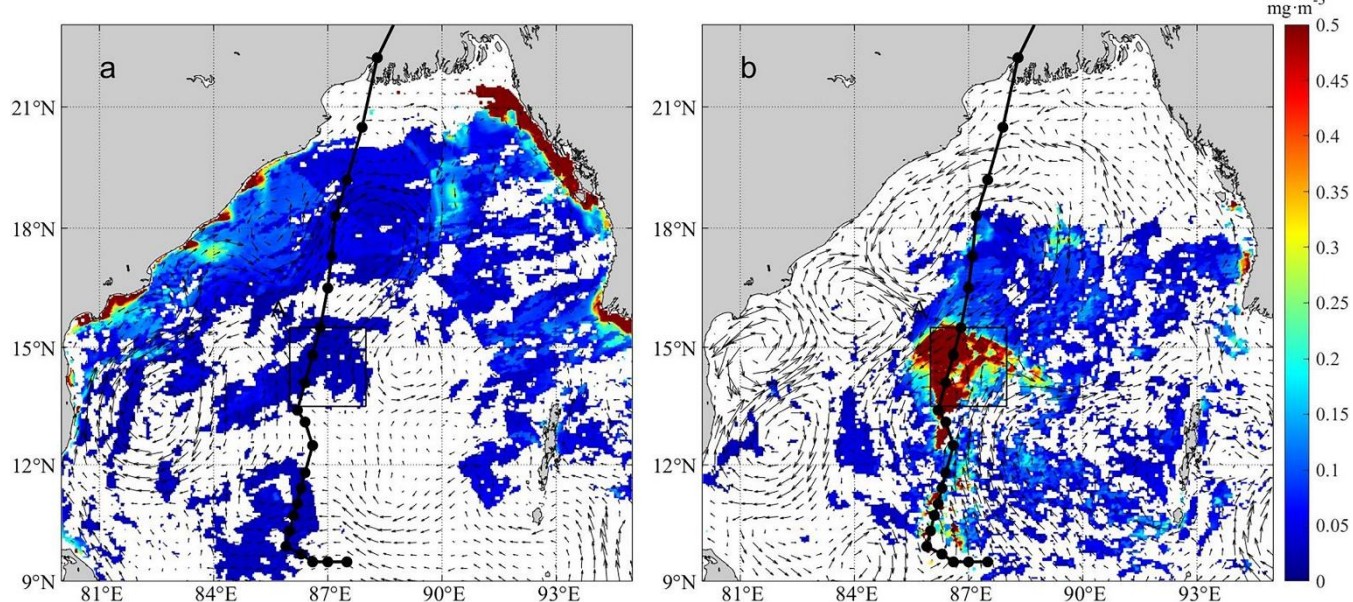

**Figure 2. Surface Chl-a concentration composited for May 12–14 (a) and May 25–27 (b). Sea surface currents are represented by black arrows.**

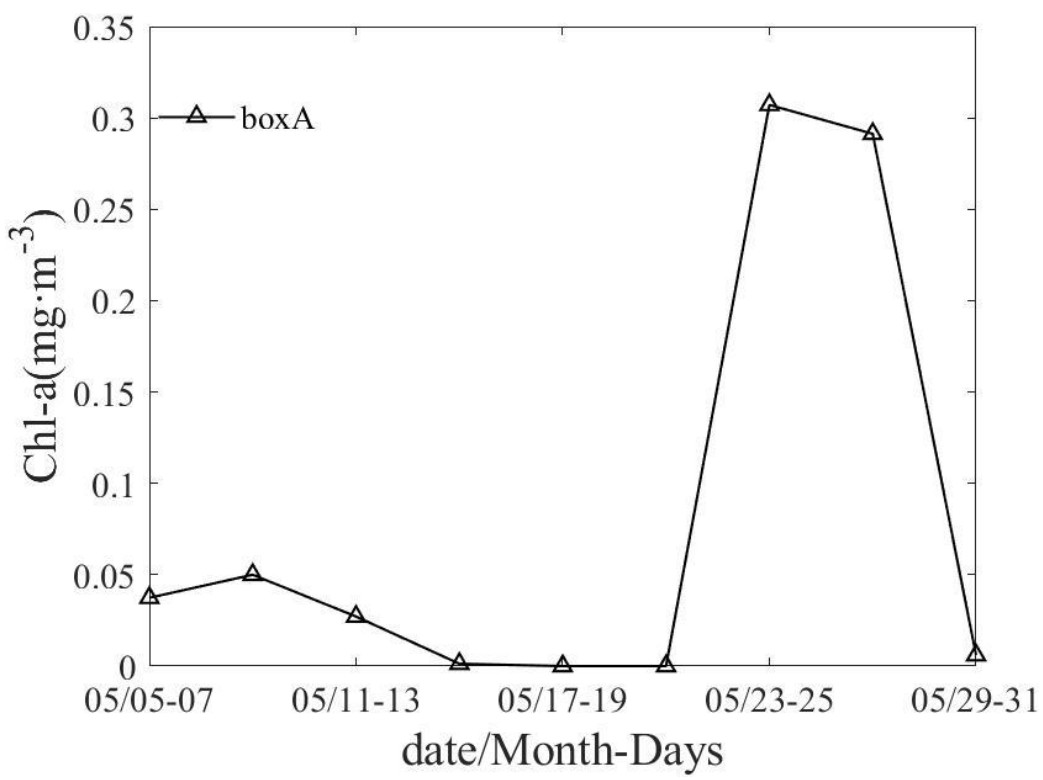

**Figure 3. Time series of three-day averaged Chl-a concentration in Box A from May 5 to 31.**

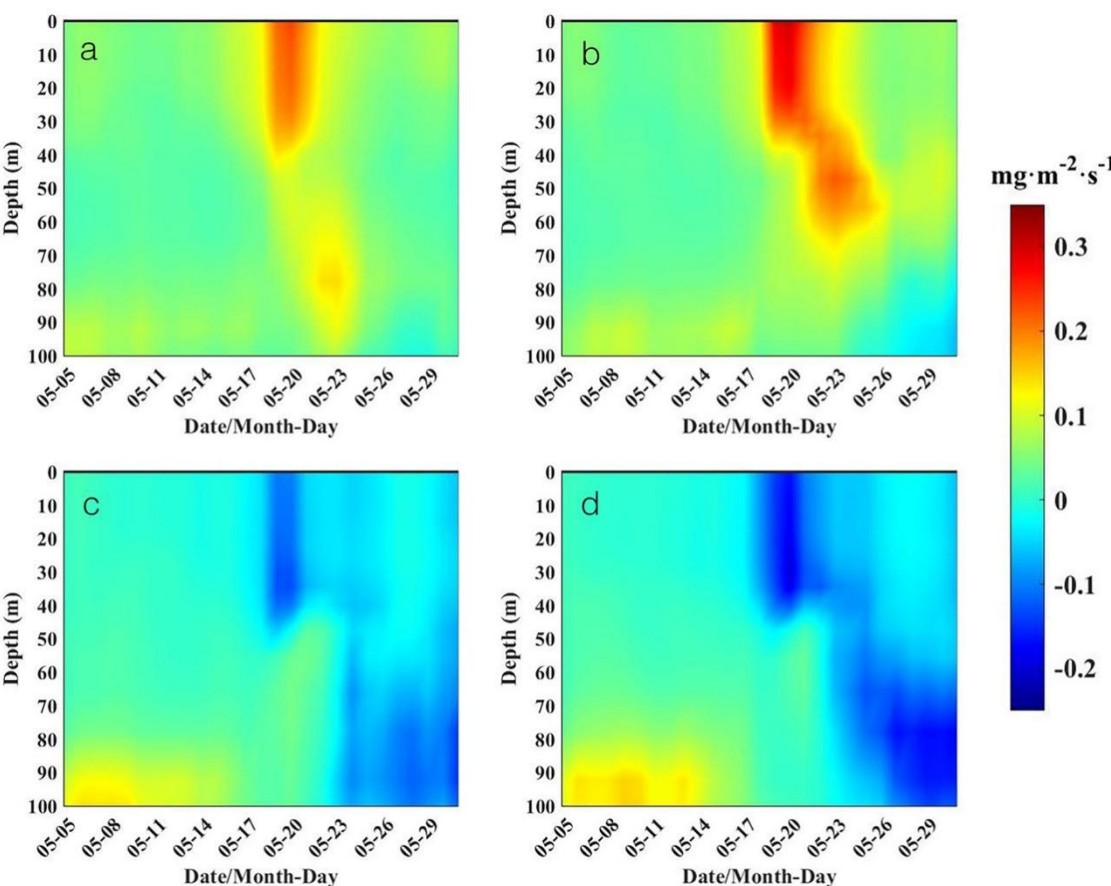

**Figure 4. Time series of Chl-a fluxes through the eastern (a), western (b), southern (c) and northern (d) sides of Box A from May 05 to 31.**

### 3.3 Sea surface height (SSH), current and temperature

The SSH decreased in the southwestern part of Box A, and the southward current increased on May 19 (Fig. 5b). Then, a cyclonic eddy lasting until three days after the phytoplankton bloom formed the next day (Fig. 5c, 5d), both of which are marked by white arrows. The role of the eddy will be discussed in Section 4.2.

The averaged TC-induced SST cooling in the Bay of Bengal was studied by Singh and Koll(Singh, 2022), who noted that the cooling temperature was 2-3 °C (0.5-1 °C) during the premonsoon (postmonsoon) season. After the passage of Amphan, the Chl-a concentration in Box A significantly increased (Fig. 3). Fig. 7 shows the vertical distributions of mean temperature from May 5 to 31 in Box A. The lowest SST was 28.6 °C in Box A after the passage of Amphan on May 20. Compared with that on May 14, the temperature decreased by 2.2 °C.

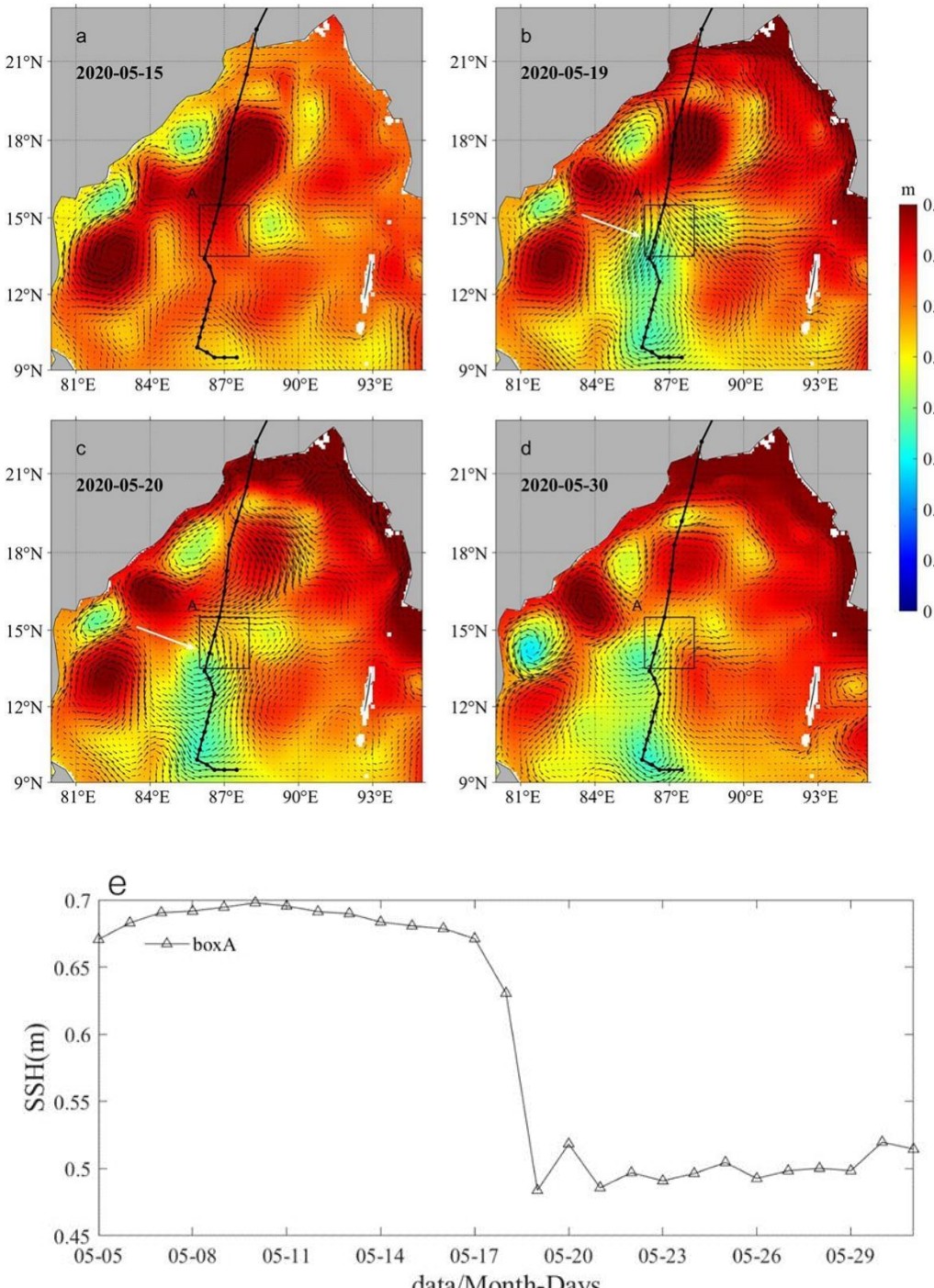

**Figure 5. Sea surface current (unit in m·s−1) on May 15 (a), May 19 (b), May 20 (c) and May 30 (d). Time series of SSH in Box A (e). The color bar represents the SSH (unit in m). The track of Amphan is marked by the black line.**

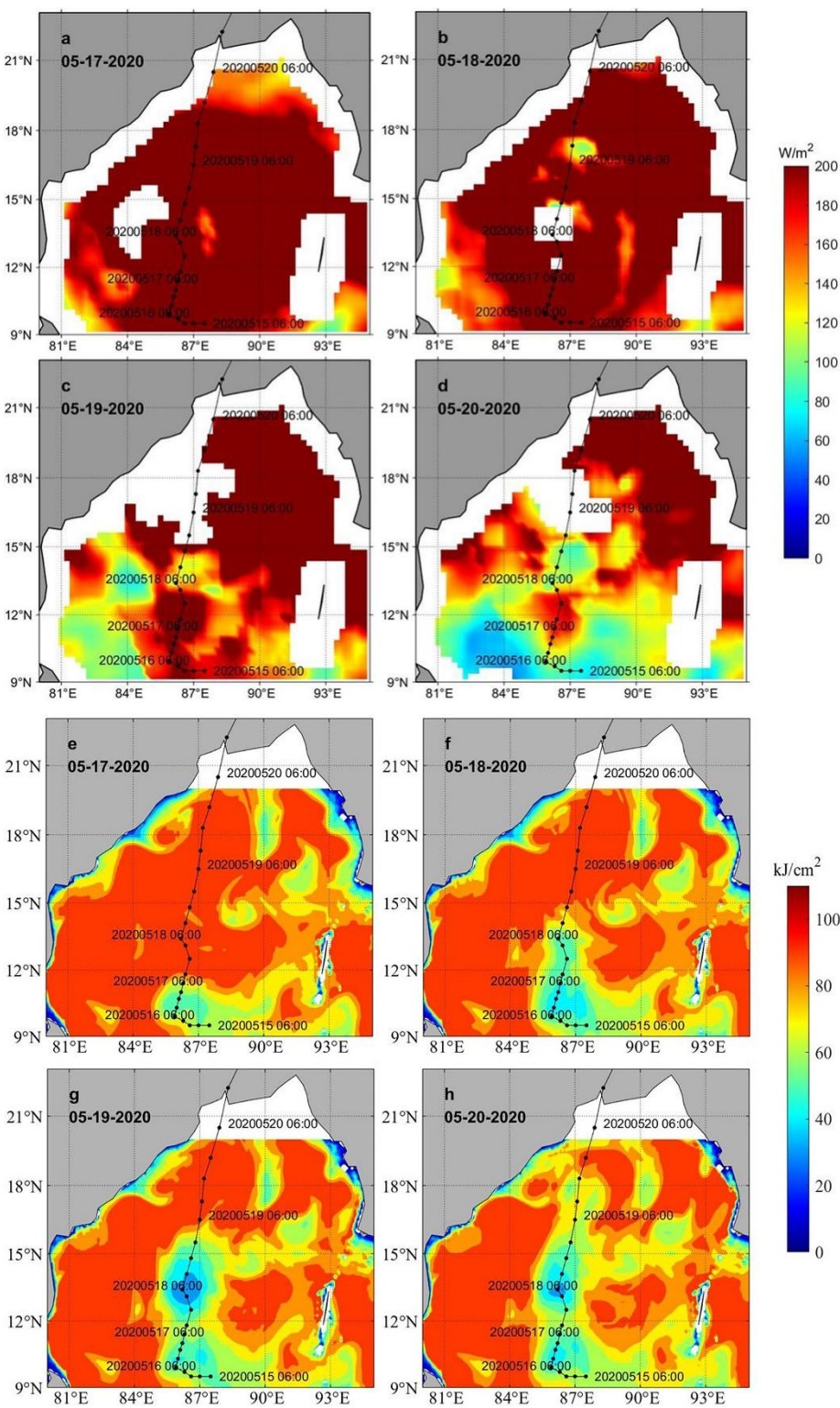

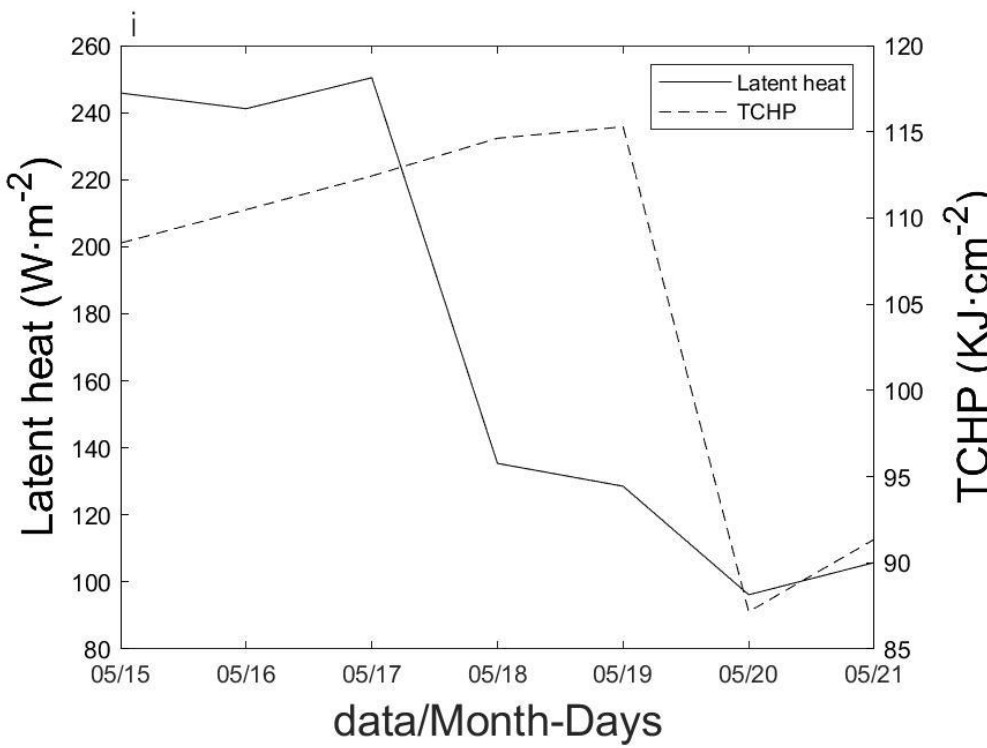

**Figure 6. The ocean heat distribution during the development of the TS. (a-d), (e-h) show the variations in latent heat flux and the changes in TCHP from 17 to 20 May 2020, respectively, and i represents time series of mean latent heat fluxes and TCHP in Box A.**

### 3.4 Latent heat flux and tropical cyclone heat potential (TCHP)

During the development of a TC, significant thermal changes will occur near the upper ocean (Liu et al., 2021). From 17 to 20  May, the average latent heat flux in Box A decreased from 250.41 to 96.19 W/m$^2$, and the TCHP had a slight rise and

175 then decreased sharply from 115.28 to 87.16 kJ/cm$^2$(Fig. 6). Moreover, as is vividly shown from the Fig. 6(i), both the latent heat flux and TCHP reached the lowest values, and after the typhoon made landfall, they rebounded. Totally,  Latent heat fluxes and TCHP correlate with changes in sea surface height because they diminish as the sea level decreases.

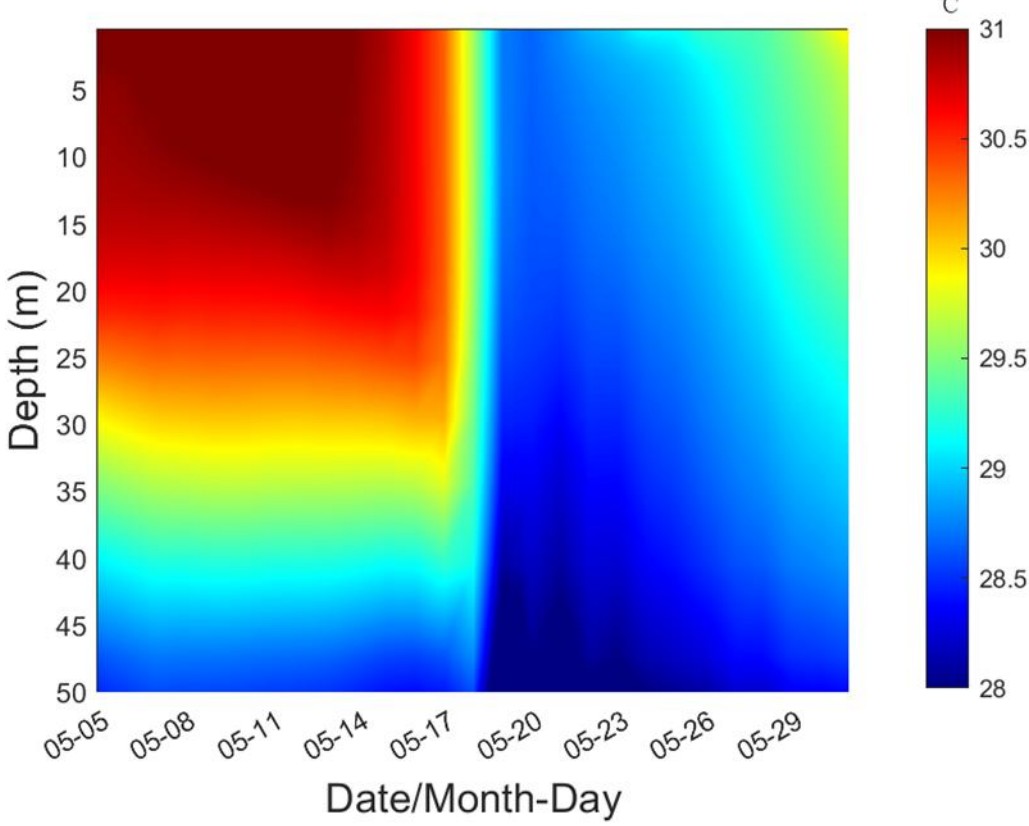

**Figure 7. Time series of vertical water temperature from May 5 to 31 in Box A.**

### 3.5 Nitrate distribution

The nitrate concentrations at different depths of z = -5 m, -40 m, -50 m and -80 m, based on the nitrate climatology data in
May, are shown in Fig. 8. Fig. 8 clearly shows that high nitrate concentrations were mainly distributed in the northeastern
and southern parts of the Bay of Bengal, and the nitrate concentration in the study area was lower, with a maximum value of
3.8 $\mu$mol·L$^{-1}$ at the depth of 80 m. Poornima et al. proposed that increased nitrate concentrations promote phytoplankton
blooms(Poornima et al., 2016) in the BoB. Although the concentration of nitrate was low in the central BoB, why did the
Chl-a bloom occur in Box A? This will be discussed in Section 4.

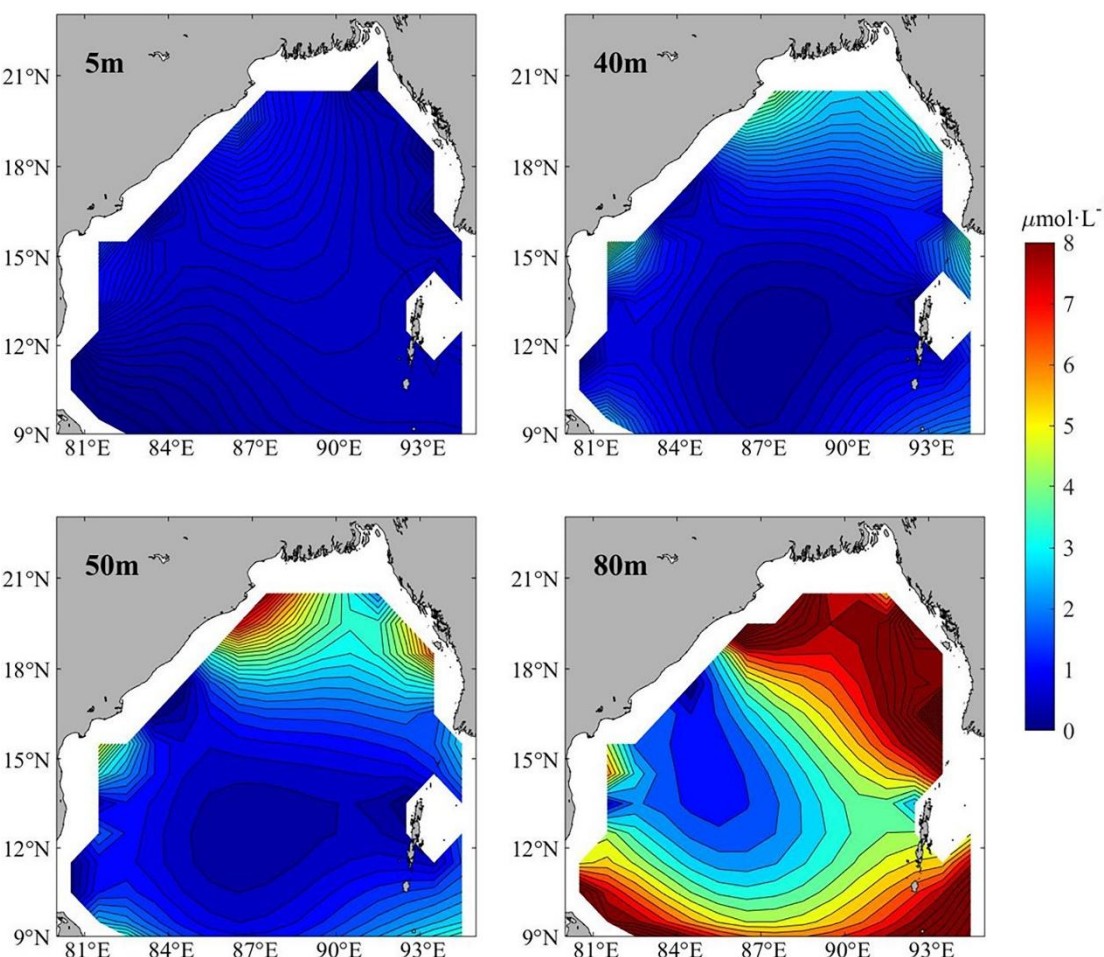

**Figure 8. Distribution of nitrate concentration (unit in μmol·L-1) in May from the WOA.**

### 3.6 Photosynthetically available radiation (PAR)

Fig. 9 shows the time series of spatially averaged PAR for Box A. During the passage of Amphan, the PAR had lower values. The cyclone began on May 15 and dissipated on May 21. The PAR remained at a high level before May 15 and rapidly dropped when the storm formed. On May 18, when Amphan arrived over the area in Box A, the PAR reached approximately zero, and then it returned to its original state after May 21, when sufficient sunlight entered the euphotic layer, which may contribute to the Chl-a bloom on May 25.

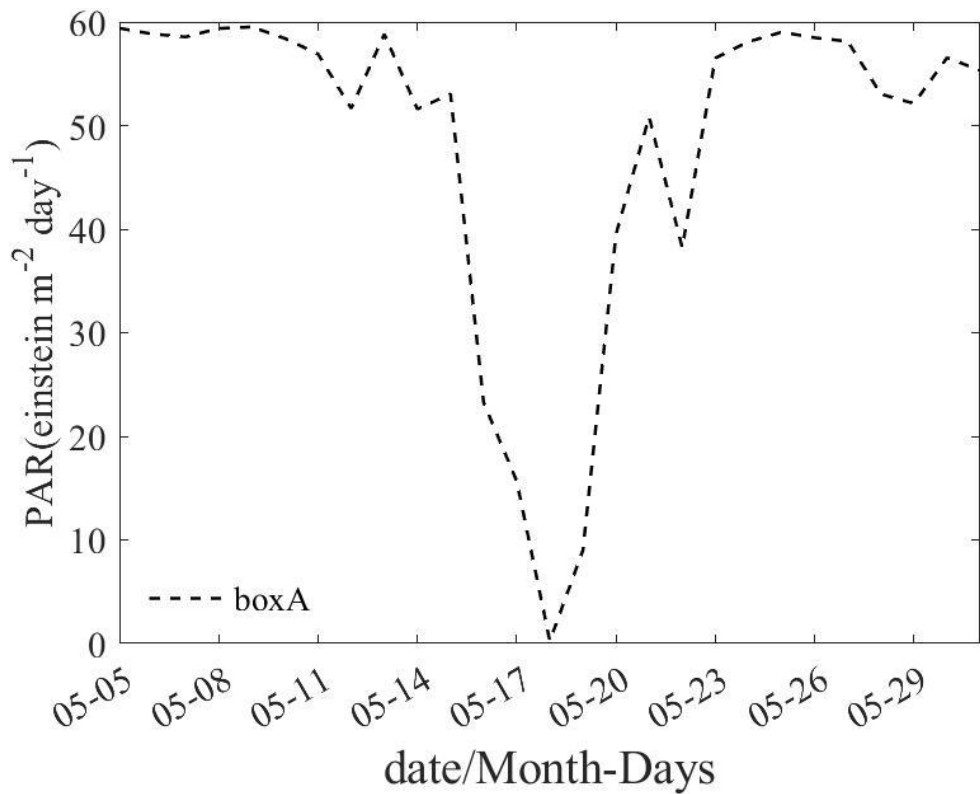

**Figure 9. Time series of spatially averaged PAR in Box A from May 5 to 31.**

## 4 Discussion

### 4.1 The effect of stratification

Generally, ocean stratification has an important effect on the Chl-a distribution in the upper ocean. The growth of
phytoplankton is inversely related to the mixing layer depth. When the mixing layer depth is shallow, an equivalent amount
of light intensity will be more helpful for photosynthesis, and when the mixed layer depth is deep, phytoplankton growth will
be limited by light intensity although sufficient nutrients may be available(Niu et al., 2016). Different BLTs in Box A on
May 8, 18, and 28 are shown in Fig. 10. The MLD deepened to 39.96 m when Amphan reached Box A (Fig. 10b), which
was conducive to Chl-a entering the upper ocean, and after the passage of Amphan, the MLD became shallower (14.91 m)
(Fig. 10c), reaching a depth where the phytoplankton could be influenced by enough PAR.

The barrier layer (BL) plays the role of a barrier in preventing vertical mixing(Balaguru et al., 2012; Sprintall and Tomczak,
1992). As shown in Fig. 10(b, c), the BLT became shallower, from 20.03 m to 5.17 m, after the passage of the cyclone,
which helped uplift Chl-a. The buoyancy frequencies of Box A on May 18 and 28 are shown in Fig. 11, showing a change
from 0.028 $s^{-1}$ to 0.025 $s^{-1}$. The weakened stratification facilitates the lifting of nutrients below the thermocline to the mixed
layer.

By means of stratification we can visualise the relationship between Chl-a flux and thermocline (Qiu et al., 2021), so we divide the upper 100 m into two layers (Fig. 12): above thermocline (0-64m) and below thermocline (64-100m). Changes in Chl-a fluxes above the thermocline was largely consistent with the overall change. The buoyancy frequency weakened to 0.025 s$^{-1}$, which favoured the entry of nutrients from below the thermocline into the mixed layer. Thus, the Chl-a fluxes below the thermocline was generally higher than that above the thermocline from 15 to 21 May.

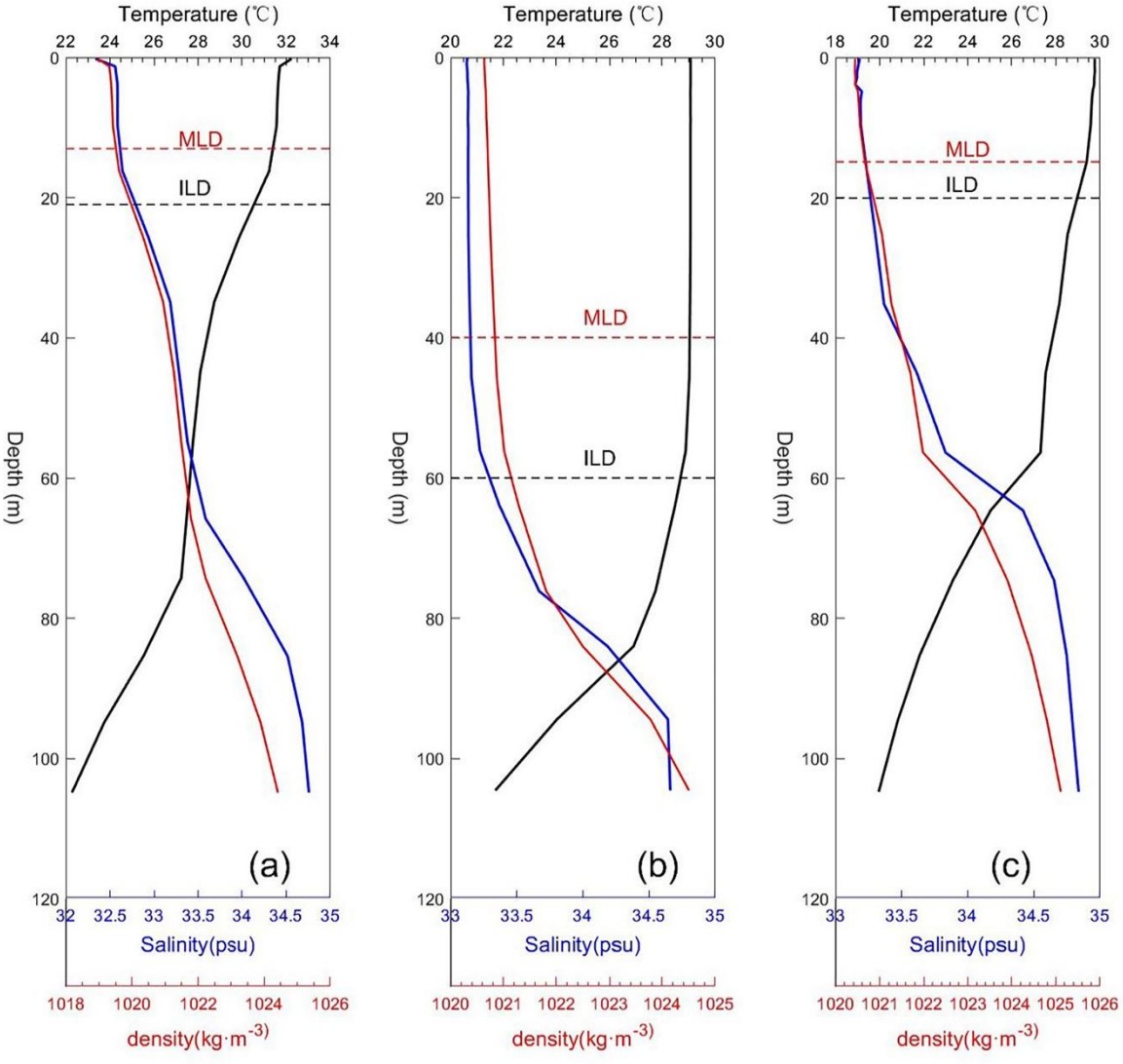

**Figure 10. Barrier layer thickness calculated from Argo data in Box A on May 8 (a), 18 (b), and 28 (c).**

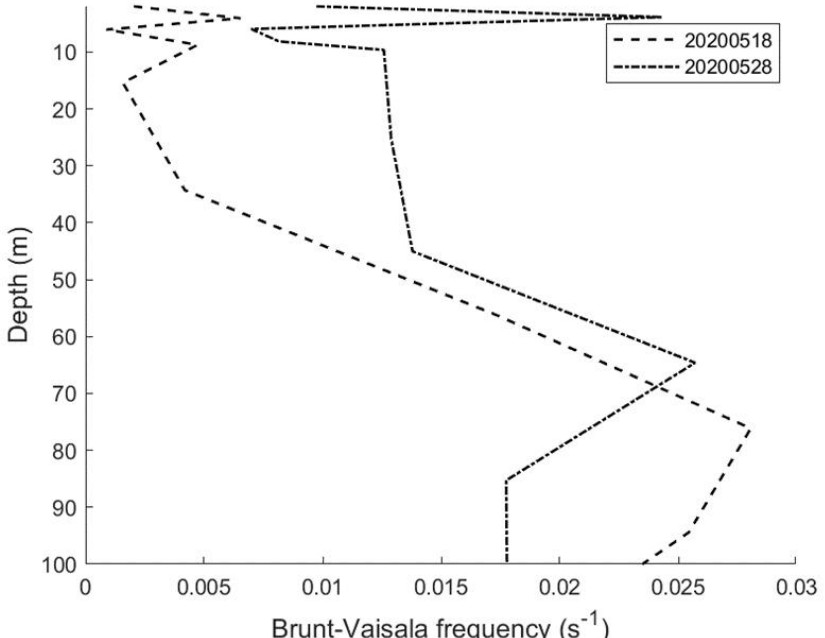

**Figure 11. Vertical profile of the buoyancy frequency in Box A from Argo data on May 18 and 28.**

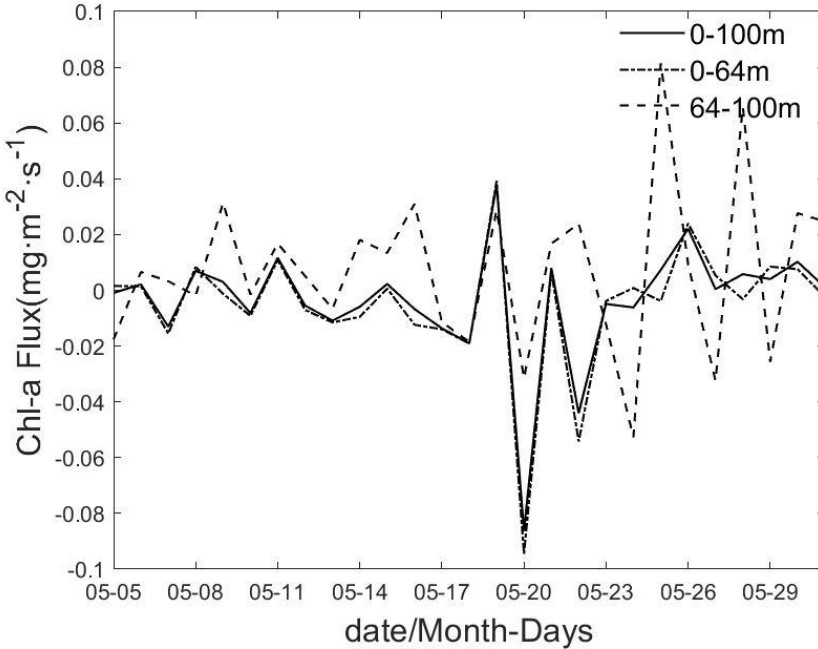

**Figure 12. Time series of Chl-a fluxes through four sides of Box A above the thermocline (0–64 m) and below the thermocline (64–100 m) and for the entire column(0–100 m).**

## 4.2 Effects of the TC and ocean cyclonic eddies

Ekman pumping appears stronger during and after the passage of TCs. Fig. 13 shows the EPV during the passage of Amphan. At 12:00 UTC on May 18, 2020, the EPV around the TC center was significantly stronger ($>5.0 \times 10^{-4}$ m·s$^{-1}$), and the strong upwelling could bring Chl-a and nitrate up to the upper ocean layer.

In order to study the effect of the TC and ocean cyclonic eddies, we quantify the intensity of the cyclonic eddies by calculating the vorticity through Equation (3). Fig. 14c shows the time series of the vertically spatially averaged negative

vorticity in Box A from May 5 to May 31. As the TC passed on May 19, an inertial oscillation with a two-day cycle period appeared in the thermocline (at a depth of 80 m) and lasted for approximately two weeks. At the same time, the latitudinal velocity also changed significantly here (Fig. 14a). This phenomenon was also found in the eastern Arabian Sea during the passage of TCs(Rao et al., 2010). During the passage of Amphan, strong vertical mixing occurred on May 19, with a maximum vorticity of 0.36 s$^{-1}$, which transported Chl-a and nitrate from the deep sea to the upper layer and accounted for the

strong upwelling of cold water (Fig. 6). Therefore, it can be proposed that during and after the passage of Amphan, deep water with rich nutrients was transported into the upper ocean through mixing, upwelling and the high PAR, triggered the surface Chl-a bloom (Fig. 8).

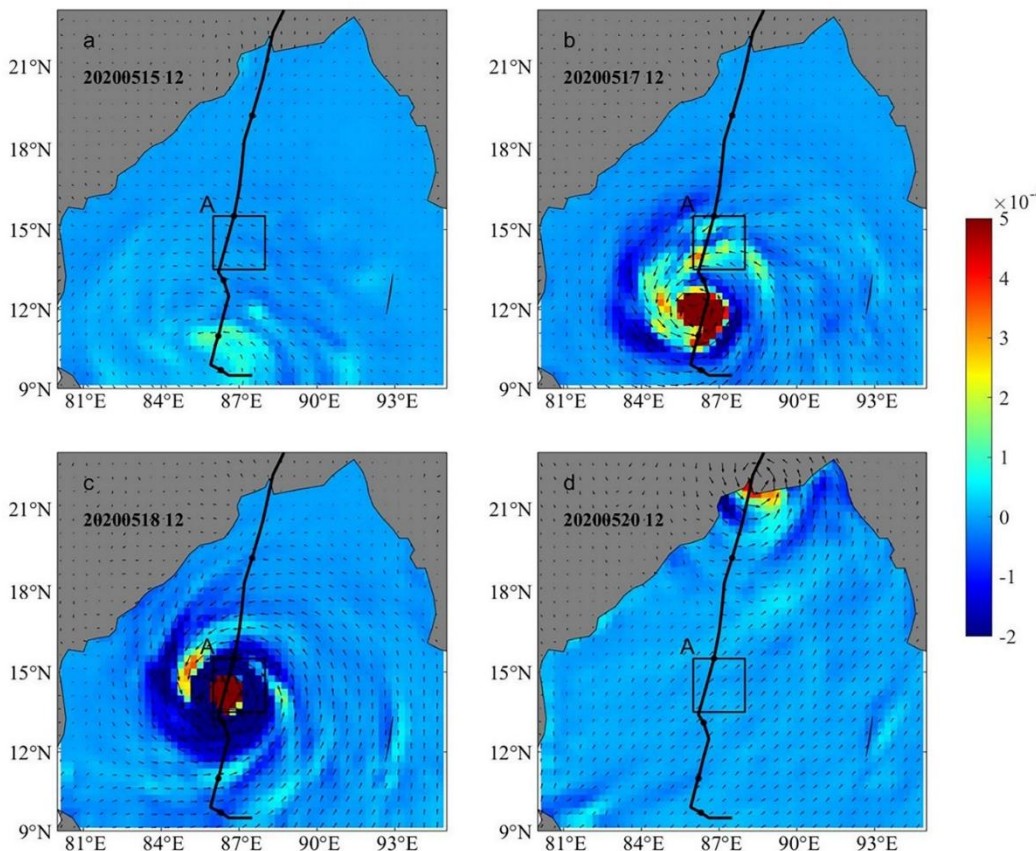

**Figure 13. Surface wind speed (arrows) and EPV (colors) during the period time of the TC.**

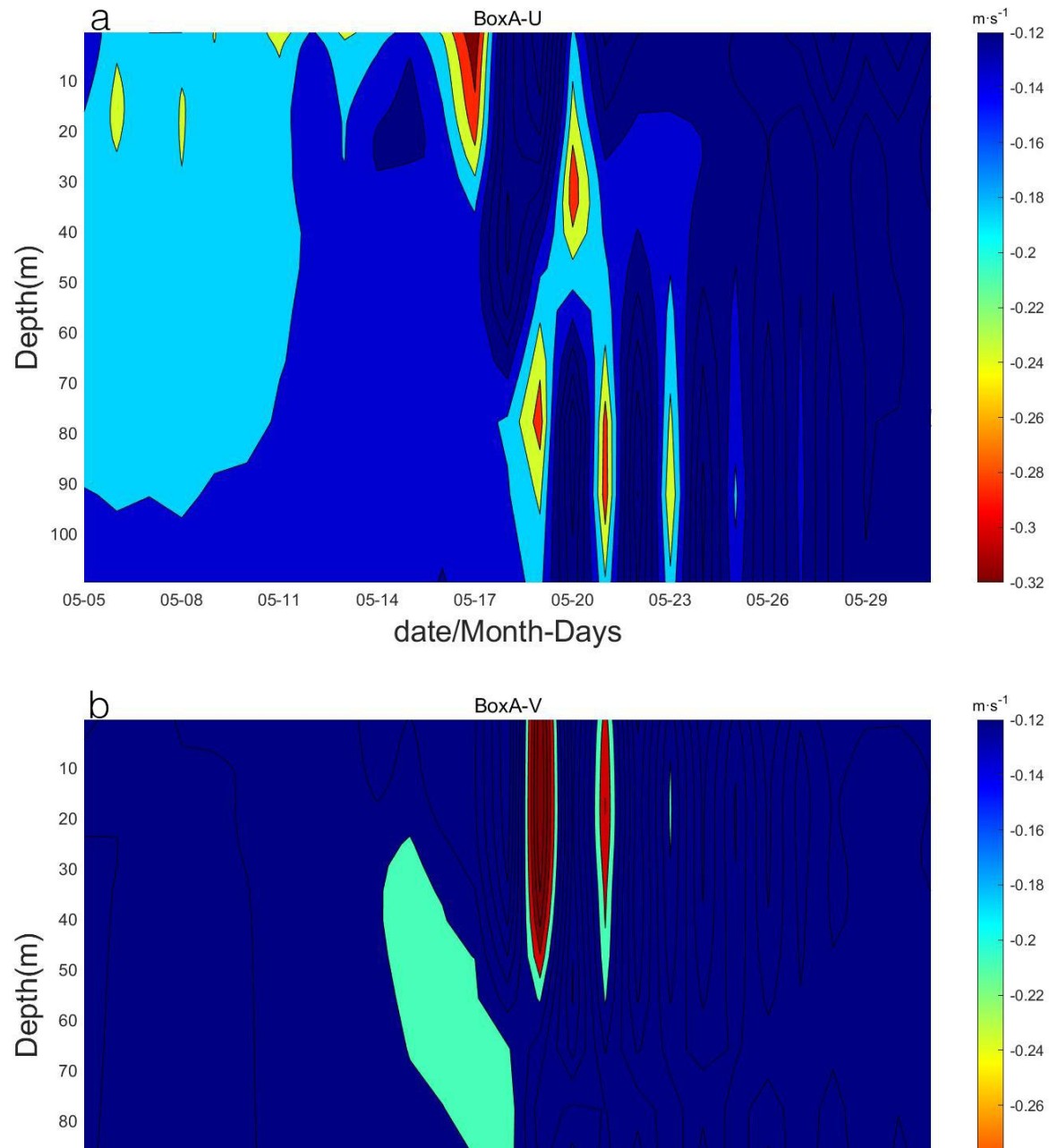

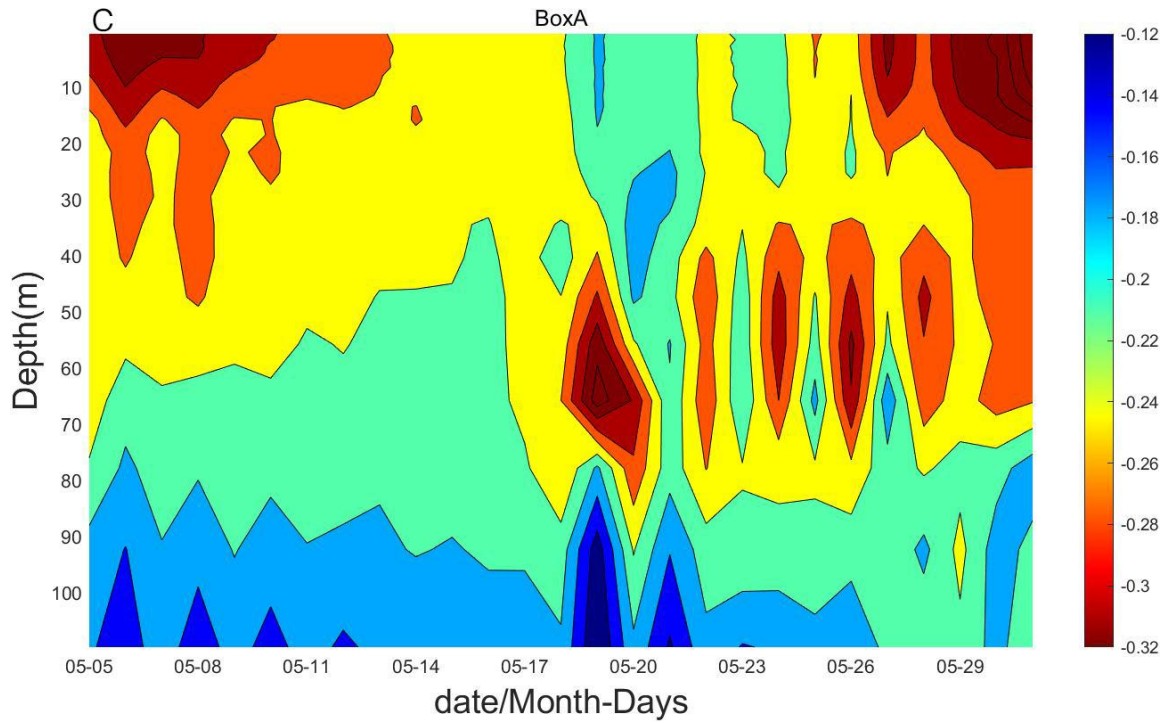

**Figure 14. Time series of latitudinal velocity (a) , longitudinal velocity(b) and spatially averaged vorticity (c) in Box A from May 5 to May 31.**

## 5 Conclusion

In this study, the dynamic and physical mechanisms of the Chl-a bloom caused by the super cyclonic storm Amphan in the central Bay of Bengal were investigated by reanalysis data, remote sensing and Argo data, which can provides new insights into the  numerical simulation of biological responses in the BoB in the future. The conclusions were as follows:

1.  In the study area, there was horizontal transport of Chl-a in the upper ocean. The maximum inlet flux speed from the western side was 0.304 mg·m$^{-2}$·s$^{-1}$, and that of the northern side reached -0.199 mg·m$^{-2}$·s$^{-1}$, which was conducive to the increase of Chl-a in Box A.

2.  After the passage of Amphan, a cyclonic eddy with a high intensity formed in the study area, and a deepened MLD, weakened thermocline and thinner BLT occurred, providing favorable upwelling conditions for the uplift of Chl-a and nitrate.

3.  With high PAR, sufficient nitrate and Chl-a in the upper layer led to the Chl-a bloom in the central Bay of Bengal.

4.  As the TC passed on May 19, an inertial oscillation with two-day cycle appeared in the thermocline (at a depth of 80 m) and lasted for approximately two weeks.

*Author contributions*. Conceptualization, Haojie Huang and Haibin LÜ; methodology, Haojie Huang, Rui Wang, Xiaoqi Ding and Haibin LÜ; validation, Hao Shen, Haojie Huang, Linfei Bai; formal analysis, Haojie Huang, Rui Wang, Xiaoqi Ding and Haibin LÜ; investigation, Hao Shen, Haojie Huang, Linfei Bai, Rui Wang and Xiaoqi Ding; data curation, Haojie Huang, Rui Wang, Linfei Bai and Xiaoqi Ding; writing—original draft preparation, Haojie Huang and Haibin LÜ; writing—review and editing, Haojie Huang and Haibin LÜ; visualization, Hao Shen and Haojie Huang; supervision, Haibin LÜ; project administration, Haibin LÜ; funding acquisition, Xiaoqi Ding, 275 Linfei Bai,Rui Wang and Haibin LÜ. All authors have read and agreed to the published version of the manuscript.

*Data availability*. The data used for this research are all from open sources listed as follows: Joint Typhoon Warning Center (https://www.metoc.navy.mil/jtwc/jtwc.html) for typhoon track data, GlobColor project (https://www.globcolour.info/)for daily Chl-a concentration, Copernicus Marine Environment Monitoring Service (http://marine.copernicus.eu/) for sea current velocity and SSH, 280 Remote Sensing (https://data.remss.com/) for SST, Cross-Calibrated Multi-Platform (CCMP, http://www.remss.com/measurements/ccmp) for the wind vector data, Indian Argo project (http://www.argodatamgt.org/accessed) for the Argos data, World Ocean Atlas (WOA, https://www.ncei.noaa.gov/products/world-ocean-atlas)for climatic nitrate data.

*Competing interests*. The authors have no competing interests to declare.

*Acknowledgements*. We thank Joint Typhoon Warning Center (https://www.metoc.navy.mil/jtwc/jtwc.html) for typhoon track data, GlobColor project (https://www.globcolour.info/)for daily Chl-a concentration, Copernicus Marine Environment Monitoring Service (http://marine.copernicus.eu/) for sea current velocity and SSH, Remote Sensing (https://data.remss.com/) for SST, Cross-Calibrated Multi-Platform (CCMP, http://www.remss.com/measurements/ccmp) for the wind vector data, Indian Argo project 290 (http://www.argodatamgt.org/accessed) for the Argos data, World Ocean Atlas (WOA, https://www.ncei.noaa.gov/products/world-ocean-atlas)for climatic nitrate data.

*Financial support*. This work was funded by the Priority Academic Program Development of Jiangsu Higher Education Institutions (PAPD), Postgraduate Research & Practice Innovation Program of Jiangsu Province (Grant no. SJCX22_1657), Postgraduate Research & 295 Practice Innovation Program of Jiangsu Ocean University(Grant no.KYCX2022-29, KYCX2022-27), Natural Science Foundation of the Jiangsu Higher Education Institutions of China(Grants no. 23KJB170005)and National Natural Science Foundation of China (Grants no. 62071207).

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
