# Peer review of "A phytoplankton bloom caused by the super cyclonic storm Amphan in the central Bay of Bengal"

_Natural Hazards and Earth System Sciences, 2023_

## Author Comment (AC1)

Dear Editor and Reviewers:

Thank you for your letter and the reviewers' comments concerning our manuscript entitled "A phytoplankton bloom caused by the super cyclonic storm Amphan in the central Bay of Bengal". Those comments are valuable and very helpful in revising and improving our paper. According to the comments, we have revised the manuscript carefully. The revisions are marked in a red font for reviewers in the revised manuscript with marked changes. Please let me know if you have any questions regarding this revision. Thank you!

Sincerely yours,

Haibin LÜ (Dr.)

2023.5.13, Lianyungang

**Response to Reviewer 1 #**

Reviewer #1: This preprint presents a valuable contribution to the understanding of the biological response of the Bay of Bengal to the passage of tropical cyclones, specifically the super cyclonic storm Amphan in May 2020. The authors have used a combination of reanalysis data, remote sensing, and Argo float data to investigate the dynamic mechanism of the chlorophyll-a bloom that occurred in the BoB during the passage of the super cyclonic storm Amphan in May 2020. The methodology is sound, the exposition is well constructed.

**Response: Thank you very much for your constructive comments! We have carefully considered all the insightful and constructive comments from you. The corresponding revisions are marked in a red font in the revised manuscript. The following is our detailed modification and explanation.**

1. However, the purpose and significance of studying this tropical cyclone were not clearly pointed out in the introduction. I think the paper requires minor revisions.

**Response: Thank you very much for your constructive suggestions! Corrected. The purpose and significance of studying this tropical cyclone has been added. (Please see lines 43-44 in the revised version)**

2. The introduction should highlight the purpose and significance of researching TC Amphan, clearly conveying why it is necessary to investigate this tropical cyclone.

**Response: Thank you very much for your constructive suggestions! Corrected. The purpose and significance of studying this tropical cyclone has been added. (Please see lines 43-44 in the revised version)**

3. Although the authors refer to inertial oscillations in their conclusion, they fail to clarify how these oscillations are linked to phytoplankton blooms.

**Response: Thank you very much for your constructive suggestions! Corrected. An article is**

cited to clarify the link between inertial oscillations and phytoplankton blooms. (Please see lines 212-214 in the revised version)

Technical Corrections:

4. What are the inlet fluxes in Line 16 and Line 226? Can you give more explanations?
**Response: Thank you for your question! We are sorry for the unclear expression. Corrected. (Please see line 16 and line 229 in the revised version).**

5. Line 11 and line 224: 'supercyclonic' should be 'super cyclonic''.
**Response: Thank you for your constructive suggestions! Corrected. (Please see line 11 and line 227 in the revised version).**

6. Figure 1: It is recommended to add variations in cyclone intensity to the figure, rather than simply a point representing the center of a tropical cyclone.
**Response: Thank you very much for your constructive suggestions! Corrected. Figure 1 has been redrawn.**

7. Line 120: 'we uses' should be 'we use'.
**Response: Thank you for your constructive suggestions! Corrected. (Please see line 121 in the revised version).**

8. Line 160: 'm·s−1' should be 'm·s−1'.
**Response: Thank you for your constructive suggestions! Corrected. (Please see line 160 in the revised version).**

9. line 174: 'μmol·L-1' should be 'μmol·L-1'
**Response: Thank you for your constructive suggestions! Corrected. (Please see lines 170 and 175 in the revised version).**

10. Figures 3,5,8,12: 'data' should be 'date'.
**Response: Thank you for your constructive suggestions! Corrected.**

11. Figure 5: The labels of figure should be enlarged to make them easier for readers to read.
**Response: Thank you very much for your constructive suggestions! Corrected. Other figures are also modified in order to make them easier to read.**

12. The abbreviations of the figures are not uniform. I suggest that unify the abbreviations for figure to 'Fig. ', like 'Fig. 1' on line 69.
**Response: Thank you very much for your constructive suggestions! Corrected. (Please see line 51、lines 121-122、line 131、lines 147-148、line 152、line 167、line 176、lines 191-192、lines 194-195、line 204、line 213、line 215 in the revised version)**

---

## Author Response (AR1)

Dear Editor and Reviewers:

Thank you for your letter and the reviewers' comments concerning our manuscript entitled "A PHYTOPLANKTON BLOOM CAUSED BY THE SUPER CYCLONIC STORM AMPHAN IN THE CENTRAL BAY OF BENGAL". Those comments are valuable and very helpful in revising and improving our paper. According to the comments, we have revised the manuscript carefully. The revisions are marked in a red font for reviewers in the revised manuscript with marked changes. Please let me know if you have any questions regarding this revision. Thank you!

Sincerely yours,

Haibin LÜ (Dr.)
2023.7.12, Lianyungang

**Response to Reviewer 2 #**

Reviewer #2: This study uses a combination of satellite, reanalysis and Argo float data to understand the mechanisms that led to the phytoplankton bloom in the central Bay of Bengal after the passage of cyclone Amphan during May 2020. The analysis shows that formation of a cyclonic eddy along the cyclone path, weak thermocline and thin barrier layer led to upwelling of the nutrients in addition to the horizontal advection of Chlorophyll to the region. The high photosynthetically available radiation in the upwelling region resulted in a phytoplankton bloom after the passage of the cyclone. The study is focused on addressing an important question relating the biological response of the ocean to tropical cyclones. The preprint is well organized and the data analysis is conducted well. The authors need to address the minor comments/revisions listed below before the preprint may be considered for publication.

**Response: Thank you very much for your constructive comments! We have carefully considered all the insightful and constructive comments from you. The corresponding revisions are marked in a red font in the revised manuscript. The following is our detailed modification and explanation.**

1. The motivation for the study needs to be more elaborated in the introduction. Why is this study important and what are the implications?

**Response: Thank you very much for your constructive suggestions! Corrected. The purpose and significance of studying this tropical cyclone has been perfected. (Please see lines 23-28, lines 43-44, line 49 in the revised version)**

2. Figure 1 does not show much information except for the track of the cyclone. The marker size can be varied to represent the strength of the cyclone along the track. The track itself can be overlaid on rainfall map with wind vectors when the cyclone made landfall. It would be useful to see the spatial extent of the cyclone.

**Response: Thank you very much for your constructive suggestions! Corrected. Figure 1 has been redrawn. Other figures are also modified in order to make them easier to read.**

3. There has to be a new figure showing the timeseries of wind stress magnitude, heat and freshwater fluxes over box A during the pre-storm and post-storm period. Highlight the dates of cyclone passage by shading in a different color. Does the PAR relate well with the shortwave radiation flux?

**Response: Thank you very much for your constructive suggestions! Corrected. A new column of values has been added to Table 1 to indicate wind stress magnitude, and new figures are added to show heat fluxes. (Please see Table 1, Figure 6 and lines 173-177 in the revised version)**

4. It is useful to mark MLD and ILD on all panels in Figure 4. It is not clearly described in the text at what depths the chlorophyl influx/outflux is occurring. This part of the discussion needs to be modified. Is it in the mixed layer or thermocline or both? Does the horizontal advection coincide with the vertical upwelling of nutrients from the deeper depths? If so, can you determine which process is more dominant?

**Response: Thank you very much for your constructive suggestions! Corrected. A new figure has been drawn to explore the relationship between Chl-a fluxes and the thermocline, and the related discussion in section 4.1 has been adapted. (Please see Figure 12 and lines 221-225 in the revised version)**

5. Figure 10 should also include a second panel with profiles of velocity shear. Do you observe increased velocity shear at the thermocline depth that coincides with weakened stratification and high Ekman pumping velocity?

**Response: Thank you very much for your constructive suggestions and your questions! Corrected. The two new figures added in Figure 14 may account for the velocity shear you propose, and Figure 14(a) Figure 14 (b) and Figure 13 clearly show that the days of strong Ekman pumping coincide with the days of high latitudinal velocity variation.**

6. There is no figure showing the inertial oscillation in the thermocline region which lasted for two weeks after cyclone passage. Vorticity in a derived quantity from the ocean velocity. Figure 12 can be a three-panel plot: Please include depth sections of zonal and meridional velocity filtered in the inertial band in the first two panels.

**Response: Thank you very much for your constructive suggestions! Corrected. Time series of latitudinal and longitudinal velocities have been presented with two new figures. (Please see lines 241-242, Fig. 14(a) and Fig. 14(b) in the revised version)**

7. The conclusions section must include a few lines on the implications and future scope of this study. For example, does this study have any implications for prediction models?

**Response: Thank you very much for your constructive suggestions! Corrected. The implications and future scope of this study has been added. (Please see lines 258-259 in the revised version)**

**Publish subject to minor revisions (review by editor)**

1. The paper is relevant for the special issue as it concerns the natural heritage through a biological response to a hazard. The editor and the referees agree on the importance of stressing the relevance of the study.

**Response: Thank you very much for your recognition.**

2. Additionally to the referee comments a geographical relevance of the study compared to other tropical ocean areas, for example Madeira/Canares or the Carribean could be done, thus increasing also the number of references.

**Response: Thank you very much for your constructive suggestions! Corrected. (Please see lines 23-28, line41,lines43-44in the revised version)**

3. The revised version after the corrections according to referee #2 should be made public before being able to meet the final decision.

**Response: Thank you very much for your constructive suggestions! Corrected. (Please see line49, lines172-177, lines221-225, lines241-242, lines258-259, Table1, Figure 6, Figure 12, Figure 14)**